# School-Based Physical Activity Interventions in Children and Adolescents: A Systematic Review

**DOI:** 10.3390/ijerph17030999

**Published:** 2020-02-05

**Authors:** Mikel Vaquero-Solís, Damián Iglesias Gallego, Miguel Ángel Tapia-Serrano, Juan J. Pulido, Pedro Antonio Sánchez-Miguel

**Affiliations:** 1Department of Didactics of Music, Plastic and Body Expression, Teacher Training College, University of Extremadura, Cáceres (Spain), 10003 Extremadura, Spainmatapiase@unex.es (M.Á.T.-S.); 2Faculty of Sport Science, University of Extremadura, Cáceres (Spain), 10003 Extremadura, Spain; jjpulido@unex.es; 3Faculty of Human Kinetics, University of Lisbon, 1499-002 Lisbon, Portugal

**Keywords:** physical activity, intervention, motivation, school based, psychosocial benefits, adolescents, children

## Abstract

*Background*: The aim of this systematic review was to examinemotivational interventions based on physical activity as precursor of psychosocial benefits inside of the scholar context. *Method:* studies were identified in seven databases (Web of Science, Sport Discuss, Scopus, Eric, Pubmed, Psycinfo and Google Scholar). The search process was from June 2011 to September 2019. A total of 41 articles met the inclusion criteria. *Results*: 23 studies showed psychological effects after intervention and also 10 studies showed psychosocial effect after the intervention. The rest of the studies, although they presented changes, did not become significant. *Conclusions*: this systematic review showed the importance of motivational processes for the performance of physical activity and sport as a precursor of psychosocial changesand highlights the importance of strategies and the temporal nature of studies to maintain significant changes over time.Likewise, the study shows the future trend of motivational interventions, highlighting the female gender as participants of special interest, and changing the methodology through web-based interventions and active breaks or mental breaks during traditional subject classes.

## 1. Introduction

The period of childhood and adolescence is a key period in the acquisition of healthy habits, since insufficient levels of physical activity (PA), during any stage of growth, can contribute greatly in the promotion of overweight and obesity [1]. In this regard, PA is understood as any body movement produced by the contraction of skeletal muscle and substantially increases energy expenditure [2]. However, there is strong evidence of the importance of PA in children and adolescents’ physical and psychological development, promoting an increase in mental acuity, acquisition of mental skills, and the adoption of more active and healthy behaviors, which help them face challenges related to higher life expectancy [3], such as those associated with low PA and increased sedentary behaviors [4].

Moreover, it is also important to emphasize some of the benefits of PAon physical, psychological, and social health. In this regard, the benefits of PA on health include the improvement of basic physical skills, physiology, morphology, body mass index, and reducing the percentage of fat [5], which is closely related to cardiovascular diseases. Also, at the psychological level, it has been associated with an increase of self-esteem [6], decrease in stress [7], emotional processes [8], cognitive achievement [9], and social behavior [10].

Hence, the volume of theoretical research on the benefits of PA currently tends to increase. Despite this increase in scientific literature [11,12,13], there are few investigations with a quasi-experimental design compared to cross-sectional designs. There are even fewer that refer to the psychosocial benefits compared to those referring to physical benefits. In this sense, the intervention process is characterized by the manipulation of independent variables over time, which provides more information about the results of intentional manipulation to improve educational outcomes. From a theoretical perspective, intervention studies help to better comprehend the causal relationships between psychological variables (i.e., motivation) and those of the educational environment [14,15], because intervention implies putting the theory into practice within the educational context. Similarly, from a practical perspective, intervention studies improve our understanding of interventions and their educational outcomes [16].

This work focuses on the role of motivation towards the intention to be physically active, adherence to the practice of PA, and the psychosocial consequences of such activity due to a healthy lifestyle, understanding as psychosocial processes those that are connected with human behavior (behavioral, attitudinal and self) in their social aspects, and that are developed over time through their interactions and experiences that allow their individual development. In this sense, motivation is an active process that encourages individuals to direct their attention and persistence towards the achievement of their goals [17]. This work has focused on two of the motivational theories that have been used most frequently in the educational context: Self-Determination Theory [18], and Achievement Goal Theory [19,20].

Self-Determination Theory [18] is the theoretical framework from which most research in this field was developed, and it has been widely used for the development of intervention strategies to improve student motivation during PA practice [21,22,23]. Furthermore, it is a macro-theory of personality and motivation, which proposes that context can influence the reasons for performing certain activities. It also focuses on the extent to which behaviors are voluntary or self-determined, maintaining that motivation is a continuum of self-determination, differentiating autonomous motivation (intrinsic, integrated, and identified regulation), controlled motivation (introjected and external regulation) and amotivation [18]. Intrinsic regulation constitutes the highest level of motivation, in which the performance of the activity in itself constitutes the objective and the gratification, producing feelings of competence and self-realization. Integrated regulation occurs when goals or regulations are fully assimilated with the self, and they are included in the individual’s self-evaluations and beliefs about personal needs. Identified regulation refers to engagement in an activity because of one’s positive assessment of it. Within controlled motivation is introjected regulation, associated with the performance of an activity to avoid feelings of guilt and to improve personal ego and/or pride [24]. External regulation refers to the performance of an activity to obtain an external reward or to avoid punishment. Finally, amotivation represents the absence of either intrinsic or extrinsic motivation [24].

According to this theory, the type of motivation shown by Individuals depends on the satisfaction of their basic psychological needs (BPN): competence, autonomy and relatedness. The satisfaction of these BPN or its lack will influence the well-being [25]. The need for competence is defined as the feeling of effectiveness while engaging in an optimally challenging task. The need for autonomy is characterized by the sense of initiative felt by the subjects when voluntarily participating in the proposed activities and, finally, the need for relatedness is defined by the individual’s feeling of integration and acceptance in the context in which he or she interacts, in this case, with other classmates, developing positive social relationships in the school context [18]. In relation to the psychosocial benefits that derive from the satisfaction of the basic psychological needs, Prentice et al. [26] suggested that they are the best defined trends to look for certain basic types of psychosocial experiences, to a somewhat variable extent between individuals, and to feel good and prosper when those basic experiences are obtained, to the same extent between individuals.

On another hand, the Achievement Goal Theory [19,20] considers that people are directed by the goals they hope to achieve and that they act rationally in accordance with these goals [27]. In this sense, the success or failure an action depends on the individual’s interpretation. Thus, what some people interpret as success is interpreted by others as failure [28]. According to this theory, achievement goals orientations refer to the cognitive representations through which students are involved in certain tasks [29]. In this line, in achievement contexts, there are two predominant types of involvement: ego-involvement, in which success is perceived when one’s performance is better than that of others; and mastery-involvement, in which the term goal implies the improvement of one’s personal competence. Therefore, the type of participation that a subject plays for a given activity will be the result of the motivational climate and the goal orientations presented by that individual [30].

Based on the above, this review aims to provide a broader view of the interventions carried out in schoolchildren, focusing not only on the benefits obtained in physical health through the promotion of PA as indicated by previous systematic reviews [31,32,33], but rather encompassing variables of a more psychosocial nature such as motivational processes, self-esteem, body image, sedentary lifestyle, quality of life, etc. This systematic review aims to answer the following questions: What incidences have the interventions based on the Self-Determination Theory or the Achievement Goal Theory on the curricular or extracurricular school context? Do PA levels increase? Do these interventions modify participants’ psychosocial behaviors such as self-esteem or body image? What is the recommended duration for the development of an intervention program? Thus, the objective of this review is to test the presence of interventions develoed in the school environment framed in the Self-Determination Theory and Achievement Goal Theory through PA or physical education have caused changes in the psychosocial aspects of children and adolescents.

## 2. Materials and Methods

This study was designed following the structure and recommendation of other systematic reviews [34,35] and the treatment used by the PRISMA guidelines for reports and studies [36].

### 2.1. Search Limits

An exhaustive search was made within the following seven databases (Web of Science, SPORTDiscuss, Scopus, Eric, PubMed (via MEDLINE), PsycINFO and Google Scholar) from June 2011 to September 2019. Moreover, this date was decided due to the existence of the last article that, according to the selection criteria, maintained a close relationship with the purposes [37]. The databases were chosen based on the articles published in journals with an impact factor in *Journal Citation Reports* (JCR). The search strategies used were developed through the combination of the following keywords for each database: (Motivation OR Intrinsic Motivation) AND (Intervention OR Programme) AND (Physical Activity OR Physical Education) AND (Children) AND (Teenegers OR Adolescents) AND (School).

### 2.2. Selection Criteria

The components of the PICOS questions were taken into account, which refer to the Population, Intervention, Comparison and Results that were answered with the aim to define the eligibility criteria shown in Table 1.

The search results were exported via e-mail to eliminate duplicates. The titles and abstracts were analyzed by two researchers independently. If there was any discrepancy, a third investigator was called to reach a consensus. Subsequently, the reference list of selected studies was reviwed to identify additional studies. The full texts of these articles were recovered. If their recovery in the databases was not possible, the author was contacted to request the full text of the research article, as it could still be in the process of being published (Figure 1).

The inclusion criteria for this study were: (1) population: school children and adolescents, age 6–18 years; (2) theoretical framework: Self-Determination Theory or Achievement Goal Theory; (3) type of study: intervention study with control group and experimental group; (4) methodology: use of PA in the intervention program;(5) main outcomes: psychosocial effects of the intervention.

The exclusion criteria were (1) population: preschoolers, adults and older people; (2) theoretical framework: Theory of Planned Behaviour, Expectancy Value Theory, and Theory of Self-efficacy; (3) type of study: cross-sectional studies, systematic reviews and meta-analyses; (4) main outcomes: studies that only show changes in biological and health indicators.

The documents included were those from important journals, which were subject to peer review. The study included articles written in English with a population of 6-18 years of age. In addition, only those that presented a longitudinal intervention or intervention design were included. Likewise, the studies carried out within the educational context were selected.

### 2.3. Data Extraction and Reliability

The content analysis was carried out in order to analyze all the information present in the articles included in this review and in relation to the previously raised research questions. Therefore, the following categories were taken into account: author/s, sample (number, age and gender), objective of the study, intervention structure, duration and frequency of the program, assessed variables, main outcomes, and conclusions.

### 2.4. Evaluation of the Quality and Level of Evidence

The criteria for assessing the quality of the studies included in the review were taken from another standardized evaluation list [38], and our selection criteria. The list was composed of three items (ABC), where criterion A refers to the subjects participating in the study, B refers to the sociodemographic characteristics and the variables that are assessed in the studies, and finally the quality criterion C, is based on the complexity of the statistical analysis used. Each item was rated with a 2 (if it fully met the criteria), 1 (if it met the minimum standards) or 0 (if it did not meet the minimum standards). For all studies, the total quality was calculated by counting the number of positive elements (a total between 0 and 6). The studies were defined as *high quality* (HQ) if they had a score of 5 to 6. A score of 3–4 determined a *medium quality* (MQ), and finally, a score of 0–2 represented *low quality* (LQ). In addition, it was decided to complement the above criteria with the assessment of whether the intervention has followed a randomized design and the reliable in its implementation.

## 3. Results

### 3.1. Overview of Studies

The total number of participants included in the present review was 14,523 participants, of which 2059 were school children between 6 and 11 years old [39] and 12,464 were adolescents between the ages of 12 and 17 years [40]. Also, a total of 245 teachers were responsible for carrying out the relevant interventions. The studies were conducted in Spain (*n* = 13), Australia (*n* = 8), United States (*n* = 5), Finland and England (*n* = 3), Canada and Italy (*n* = 2), South Korea (*n* = 2) Norway, Poland, Portugal, Estonia, Malaysia and Thailand (*n* = 1). Table 2 provides an overview of each of the 33 articles that contributed the data of this review.

### 3.2. Educational Level Uses

The most frequent grade levels for the performance of interventions based on the Self-Determination Theory and the Achievement Goal Theory in PA is Secondary Education (11–17 years), with 40/45, and to a lesser extent in Primary Education (6–11 years), with 5/45.

### 3.3. Design of the Programs

Most of the interventions in this review 39/45 presented a quasi-experimental design (2 × 2), a control group (CG), and an experimental group (GE) with two collection measures, one at the beginning of the program and another at the end of it [39,76]. Other investigations presented works of a different nature 6/45, where the same group was assessed at the beginning, and another one at the end of the intervention program [46,65,78]. Likewise, intervention processes with quasi-experimental design, a CG and an EG were also found, and three data collection moments 5/45 [47,62,63]. In this regard, researches were also found with four data collection moments 2/45 [22,54,68]. Finally, we might highlight a study where data was collected from the experimental group at three different times (i.e, at the beginning of the program, in the midle, and the follow up measure), and the control group only the initial measure and the follow up [42] (Table 3).

### 3.4. Nature of the Variables

All the studies present in the current review assessed psychosocial variables. In this sense, motivation is the axis on which changes in the different variables were based. Thus, 43/45 studies of the present review assess motivation according to its type: autonomous, controlled, and amotivated, or through satisfaction, thwarting, or support of BPNs. Accordingly, this involved assessing support by teachers, parents, and peers in 16/45 studies [39,42,53,73]. Also, the presence of variables such as enjoyment and boredom is common in this type of studies 13/45 [42,50,65,74,76]. Similarly, the assessment of sedentary behavior and the intention to be physically active is quite frequent in these investigations 8/45 [43,50,61,62,63,73]. To a lesser extent, other variables are present, such as: predisposition to contentedness [76]; quality of life [40,69]; well-being [77]; screen time [64,78]; diet and nutrition [52,59]; self-efficacy [39,71]; cooperation, affect, satisfaction, effort, psichologic condition, Perceived behavioral control, body image, and utility [40,42,44,49,57,61,67]. On another hand, more physical and anthropometric variables have also been considered. In this sense, 22/45 studies took into account PA [40,51,54,64,68], and 9/45 used anthropometric measurements [40,59,63,64].

### 3.5. Outcomes

Three research questions were asked from the 45 studies:What incidence do interventions based on Self-Determination Theory or Achievement Goal Theory have in the curricular or extracurricular school context?Do PA levels increase? Do the participants’ psychosocial behaviors modify their self-perception, body image, levels of motivation, quality of life, fun, boredom, and the psychosocial climate?What is the recommended duration for the development of an intervention program?

### 3.6. Focus of Studies and Context

All the studies of the review presented the application of a PA program in the school environment based on two of the most used theoretical models in the study of motivation: Self-Determination Theory [24]; and Achievement Goal Theory [19,20]. In this sense, a large number of investigations of this review were based on the postulates of the Self-Determination Theory 42/45, of which 8 of them were based on both theoretical frameworks (Self-Determination Theory and Achievement Goal Theory), and 11/45 of the total of works were based on the Achievement Goal Theory. Only one study was completely based on the Achievement Goal Theory [45]. In two of the studies based on the Self-Determination Theory [51,59], the Cognitive Social Theory [81] was also used to assess perceived competence.

### 3.7. Outcome on Motivation

Table 4 shows in more detail the results obtained in the different interventions that are part of this systematic review for each variable, showing the results for the different conditions of the participants (control and experimental). In this sense, the present review shows changes in the levels of motivation in 45/45 of the selected articles. In view of the totality of concepts that included in the two theories, it would be necessary to specify in which dimensions these changes take place. In this line, 14/45 (31.1%) of the studies showed changes in autonomous motivation [67,68,78] but not in all case these changes have been positive or significant [52,73]. In this regard, the motivational construct has not always been directed for PA [74], but also toward caloric intake [52], and screen time [77]. Similarly, other studies 14/45 (31.1%) of these investigations had positive effects on intrinsic motivation [39,42,64,75], 14/45 revealed effects on identified regulation (31.1%) [39,42,48,64,76], and 2/45 (4.4%) of the investigations showed effects on integrated regulation [54,61].

In relation to controlled motivation 2/45 investigations (4.4%) showed an increase in scores after the application of the intervention program [59,63]. Regarding the most controlled forms of regulation according to the SDT, introjected regulation [39,49,64]. On another hand, 5/41 (12.1%) of the investigations revealed effects on external regulation, reducing it in some cases [22,42,65], and increasing amotivation in others [59,64].

In relation to the effects produced in the BPN, 21/45 (46.6%) of the investigations showed effects after the application of the interventions. In this line, we should differentiate these effects according to the nature of the type of BPN (satisfaction, frustration and support). 13/45 (28.8%) of the studies showed effects on BPN satisfaction [50,76], 7/45 (15.5) effects on BPN thwarting [22,47], and 14/45 (31.1%) support on BPN [73,77], showing an increase in the satisfaction of autonomy in the experimental group. Likewise, the studies by Amado et al., Bechter et al. [42,44], and Cheon et al. [47] revealed an improvement in autonomy, competence and relatedness after the intervention. Other studies [74,76] showed positive effects of the interventions on the variables of satisfaction of autonomy and competence. Abos et al. [41] AndGonzález-Cutre et al. [54] reported a significant improvement in the experimental group in satisfaction of the three BPN. On another hand, studies revealed effects related to the thwarting of BPN. In this sense, the intervention of Cuevas et al. [48] increased BPN thwarting, whereas other investigations reported a decrease in those scores [22,47].

Regarding the effects in the studies based on the Achievement Goal Theory, 6/45 (13.3%), in all of them, there was an increase in mastery involvementtask-oriented climate [41,60,72,76], and in 6/45 (13.3%), there was a decrease in ego orientation [45,55]. On the other hand, there was an intervention that had no effect between pre and post, and also showed a reduction in the task-oriented climate and an increase in the ego-oriented climate [56].

### 3.8. Outcome on Physical and Physiological Behaviors

A total of 33/45 (48.9%) studies assessed the effects on physical behaviors through variables such as: the intention to be physically active 7/45 (15.5%), PA 21/45 (46.6%), and sedentary lifestyle 3/45 (6.6%), attitude 12/45 (26.6%), effort 4/45 (8.8%), control of the perceived behavior 2/45 (4.4%) and commitment 1/45 (2.2%). Thus, in relation to the intention to be physically active, most of the studies reported small changes that did not reach statistically significance [48,50]. However, the study performed by [41] revealed significant changes for the predisposition towards PA after the intervention. Likewise, most of the intervention studies showed a statistically significant increase in PA scores [51,54,55,66,79]. However, some intervention studies did not show significant changes in the variables tested [46,56,60,65,68,69]. Similarly, in relation to sedentary lifestyle, some interventions [63,64] managed to reduce it significantly, although other investigations [52] did not find significant changes.Regarding the attitude generated by the interventions among its participants different types are showed: affective, instrumental, competitive, towards the contents of PE and perceived benefits [41,57,70,71]. Moreover, other interventions experienced effects on the effort [44], perceived behavior [22], and commitment [23].

On another hand, 6/45 studies (13.3%) included physiological and anthropometric variables such as adiposity, body mass index, and cardiorespiratory fitness. In this line, regarding the body mass index, some studies of the present review showed that, although it was reduced in some of the cases [39,40,59,64,78], this decrease did not reach significance. No changes were revealed in the adiposity index [64]. Finally, there was a significant increase in cardiorespiratory fitness [40,63,77].

### 3.9. Outcome on Psychosocial Effect

A total of 46.6% of the studies (21/45) have addressed psychosocial variables such as body image, quality of life, wellbeing, fun, boredom, usefulness, self-efficacy, appearance, psychological condition, perceived barriers, subjective norms, behavioral perceived control, psychosocial climate, social support, and prosocial and antisocial behavior. In this line, 10/41 studies (17.1%) revealed effects on quality of life [40,69], body image [40,57], self-concept [54], wellbeing [43], self-efficacy [41], appearance [57], psychological condition [57], perceived barriers [71], subjective norms [54], behavioral perceived control [61], and prosocial behaviors [23,47,61,71,80], and psycho-social climate [45]. Similarly, 10/45 studies (30.3%) revealed positive effects on fun along with boredom reduction [42,48,50,51,65,74,76]. However, other studies showed minor effects on enjoyment. Also, an increase in the perception of utility after the application of the intervention program [39,42,46,66]. On another hand, in Rhodes [70] intervention program based on autonomous motivation, the perception of utility decreased.

### 3.10. Outcome on Duration of the Program

The duration of the interventions ranged from four weeks [66] to two years [56,60], but most of the intervention programs had a duration that ranged between 5 and 20 weeks [40,64]. Likewise, the number of lessons was between eight sessions [76] and 24 sessions [43,50]. Similarly, the interventions with the greatest effect were those whose duration ranged between three months and one year 19/45 (42.2%) [78]. Moreover, as mentioned in the inclusion criteria, duration was not a determinant for the elimination of the article. In this sense, 19/45 (42.2%) of the interventions performed lasted less than three months and did not show significant effects [41,42,45,46,48,51,66,67,68,73,76,77,79]. However, it is important to point out five of them, which obtained significant results [46,51,61,72,77]. This criterion, which evaluates the effect of the intervention according to its temporal nature, has been used in a review of a similar nature [82]. On another hand, 6/45 (13.3%) studies [55,56,60,62,63,69] presented a duration of more than one year, but the effects of intervention did not show changes in some of the collection moments.

### 3.11. Main Effect on the Intervention

The effect sizes for each variable of each of the interventions included in this systematic review are described below (see Appendix A).

Not all articles in this review have effect sizes, in some cases they do not report such indormation, or the effects are not strong enough to be categorized as small. In addition, it should be taken into account that not all studies have the effect size with the same statistic (Cohen’s *d*; *η*^2^; *η*^2^*p*; Clif delta; R^2^).

Regarding the motiation construct, Bechter et al. [44], andRiiser et al. [40] showed showed a small effect after his intervention, according to Cohen’s *d* = 0.11. Cuevas et al. [48], Piipariet al. [68], and Sánchez-Oliva et al. [73] showed average effect size *η*^2^ > 0.06. Similarly, with regard to the different autonomous motivation regulations, specifically intrinsic regulation, Nicaise et al. [65] showed after its intervention an insignificant effect size, Cohen’s *d* < 0.11. In this sense, Fernández-Rio et al. [49], and Sevil et al. [75] showed a small size effect Cohen’s *d* = 0.20, Lubans et al. [64] had a médium size effect (Cohen’s *d* = 0.53). González-Cutre et al. [50], Sevil et al. [74], and Sevil et al. [76] revealed high size effects. Regarding integrated regulation, González-Cutre et al. [50] showed after its intervention an average effect size (Cliff delta = 0.36). Regarding the identified regulation, Fernandez-Río et al. [49], Cuevas et al. [48], Lubans et al. [64], andNicaise et al. [44] showed an increase with a small effect size. Amado et al. [21], Sevil et al. [75], and Sevil et al. [76], showed after their interventions an average effect size. Finally, González-Cutre et al. [54], and González Cutre et al. [53] showed an important effect after the application of their interventions. Regarding the most controlled forms of motivation, studies of Gonzalez-Cutre et al. [53], andLubans et al. [64] showed an increase in introjected regulation with an average effect size. Regarding external regulation, Fernandez-Rio et al. [49], González-Cutre et al. [53], andLubans et al. [64] revealed a reduction in their work with small and medium effect sizes respectively. Finally, Sánchez-Oliva et al. [73] demonstrated a decrease in controlled motivation with an average effect size.

In relation to the effects on BPN we have to distinguish between satisfaction, support and twarting. With respect to the satisfaction, Amado et al. [21], Lonsdale et al. [62], How et al. [58], and Sánchez Oliva et al. [73] reported a small effect on autonomy satisfaction. Consistent with this result, Cheon [22,47] they showed, respectively, a positive effect of a moderate nature in the satisfaction and support of autonomy, and in the satisfaction of the threeBPN, while the frustrations of the 3 BPN are reduced.Similarly, Tilga et al. [80], showed changes in the satisfaction of the BPN while reducing the thwarting of the BPN of autonomy with small effects. Franco and Coteron [50] and Sevil et al. [76] reveal important effects on the satisfaction of autonomy and competence. Moreover, Bechter et al. [44] showed after his intervention moderate and small effects on autonomy, competence and relatedness, respectively. Laroche et al. [61]. Lonsdale et al. [62] and Perlman et al. [67] demonstrated, after their interventions, a small-moderate effect on the satisfaction of the BPN. Regarding the support of the BPN, García-Calvo et al. [23], González-Cutre et al. [53], González-Cutre et al. [54], Piipari et al. [68], Sevil et al. [75], Sevil et al. [76] and Shannon et al. [77], showed a moderate effect of their interventions. In this regard, Sánchez-oliva et al. [73] emphasized a small effect on autonomy support (*η*^2^*p* = 0.08) and relatedness (*η*^2^*p* = 0.05). Finally, respecting the thwarting of the BPN, Cuevas et al. [48] showed a reduction in the need for thwarting for the experimental group with an effect size (*η*^2^ = 0.037)

Regarding the effects shown according to motivational orientations, Kokkonen intervention works et al. [60] and Rokka et al. [71] showed small size effects (*η*^2^ = 0.04) in the task-orientated climate. Similarly, Grasten et al. [55], and Sevil et al. [75] reported average effect sizes (R^2^ = 0.37). Finally, Bortoli et al. [45], and Sevil et al. [76] obtained important size effects on the task-orientated motivation. (Cohen’s *d* = 0.80). On the other hand, respecting the performance motivational climate, Grasten et al. [55], Kokkonen et al. [60], and Sevil et al. [76] reported after their interventions a small effect size (*η*^2^*p* = 0.036). In this sense, Bortoli et al. [45], Rokka et al. [71], and Sevil et al. [75], showed a reduction in ego-oriented scores with medium effect sizes (Cohen’s *d* = 0.50).

Concerning the effects produced in the behaviors, most of the intervention studies had a small effect (Cliff’s delta = 0.22) [50,73,76] in the intention to be physically active. In accordance to this, the study conducted by Cuevas et al. [48], and Rhodes et al. [70] presented an average effect size (*η*^2^ = 0.015), and finally some medium important effect sizes were shown in Abos et al. [41], and González-Cutre [53] (Cliff’s delta = 0.45). Regarding sedentary behavior most of the studies [47,48,62] showed a small-medium effect size (Cohen´s *d* = 0.13), and onlyLubans et al. [64] showed an important effect size (Cohen’s *d* > 0.80). In relation to studies that improve the quality of life, Girelli et al. showed average effect sizes. Respecting the improvement in PA, most of the studies reported small-medium effect sizes [40,46,54,55,56,58,62], only two studies showed important effects on physical activity [53,63]. With regard to the attitude, this is treated differently by various authors. On the one hand, in relation to the attitude as a general dimension, small average effect sizes were shown (Cliff delta = 0.34), in the cognitive attitude, [41,74] important size effects were shown, and in the attitude towards learning [49] and competition [57] médium size effects were shown. Finally, respecting the effort [44,72] a médium size effect was shown (Cohen’s *d* = 0.30).

Regarding anthropometric variables and cardio–respiratory fitness, most intervention studies did not report effect sizes. In this regard, Lubans et al. [64] revealed an important size effect on cardiorespiratory fitness (Cohen’s *d* = 5.9), andRiiser et al. [40] showed an important size effect on body mass index (Cohen’s *d* = −0.70).

Regarding the effects on psychosocial variables, it should be noted that the intervention de Riiser et al. [40] showed effects medium effects on body image (Cohen’s *d* = 0.56). Furthermore, respecting appearance, Hajar et al. [57] showed after their intervention a médium size effect. Moreover, regarding quality of life, Riiser et al. [40], and Shanon et al. [77] showed small and medium effect sizes respectively. In this sense, Babic et al. [43] showed a small effect size on well-being. Regarding the impact of interventions on fun and boredom, most studies [41,50,57,67,74,75,76] showed medium-important effect sizes in the fun. On the other hand, the interventions of Cuevas et al. [48], and Sevil et al. [74] reduced boredom scores with average effect sizes. Respecting self-efficacy, the researches [44,46,76,80] showed a medium-sized effect after the application of the intervention. With regard to the psychological condition, the intervention of Hajar et al. [57] showed a small-medium impact on the effect size (*η*^2^ = 0.09). Finally, in relation to the impact on prosocial and antisocial behavior, important studies [45,47,53,61,76] showed medium-important effect sizes regarding the promotion of prosocial behavior. In this line, the interventions conducted by Bortoli et al. [45], and Cheon et al. [47] showed a reduction in boredom with an average effect size.

## 4. Discussion

The objective of this article was to review intervention studies based on the application of Self-Determination Theory, Achievement Goal Theory, or both carried out in the educational context. For this purpose, only intervention programs that present evidence of methodological quality were included. The results and their implications have been ordered according to the previously presented research questions.

Thus, in relation to the first question, the findings showed that most of the intervention studies are based on the Self-Determination Theory [18]. This may be due to the amount of constructs concerning human behavior that are included. In this sense, Self-Determination Theory itself in its postulates emphasizes that many of the investigations based on this theory have assessed environmental, social and personal factors in order to assess their effect on the behavioral manifestations of the individuals [24]. In this regard, many of the intervention programs provide instructions to the teacher as a means of transmitting techniques that increase or decrease levels of motivation.On another hand, some of the interventions were developed from both motivational theories [60,63,76], the Achievement Goal Theory and the Self-Determination Theory. In this line, previous studies show that these two methodological approaches can complement each other to give a more complete view of motivation independently of an educational or sporting context.In this sense, Lazowsky andHulleman [16], showed in their review and meta-analysis that a large part of the interventions developed in the educational field used a multiple theoretical perspective (two or more different theories). Finally, this review revealed that only three of the interventions was based solely on the Achievement Goal Theory [19,20]. In this regard, Bortoli et al. [45] reported in their study that, after the application of the intervention program, a task-orientation was more strongly related to positive emotional states as opposed to ego-directed intervention. All articles presented greater or lesser effects on motivation, satisfaction and support of the needs of autonomy, competence, and relatedness through the application of techniques and strategies taught by teachers or researchers [42,54,73,77]. In this sense, this review showed that interventions carried out in the educational context have an important medium effect on the most autonomous forms of motivation [44,48]. This fact may be due to the fact that high scores are reflected in the most intrinsic forms of motivation, and that the inclusion of innovative elements as support through the Internet [40] or the use of exergames [70] can easily manipulate these types of motivation. Likewise, it was shown that the changes in motivation are mostly predicted by the support and satisfaction of the BPN [22,47,54], which showed moderate exchange effects. This may be due to the fact that in many of the interventions they base their strategies on the support of autonomy [58,74], relegating to a second level the basic psychological need for competence. In this regard, this review has shown that satisfaction or support directly affects the motivation generated (autonomous or controlled) [50,67,76] and the goal orientations [74].

The intervention programs presented in this review also revealed effects on fun, boredom, and usefulness for the improvement in levels of motivation and PA in the educational context. In this line, Franco and Coteron [50] pointed out that several studies agreed on the positive effects in supporting autonomy or other BPN. Likewise, Rokka et al. [72], in their intervention developed around the TARGET areas [19] and support for autonomy, reported an increase in motivation and fun [19]. Showed that ego-involvement motivation would have a greater effect on boredom. In accordance to this issue, Texeira et al. [37] demonstrated that the most autonomous forms of behavioral regulation and PA can be important in the context of promoting an active lifestyle through the choice of optimal challenges that promote compromise and enjoyment. Also, some of the studies showed effects on the improvement of body image, and quality of life. In this regard, quality of life improved in the studies of Quaresma et al. [69] and Riiser et al. [40], associated, on the one hand, with the support of parents and partner, and, on the other, with body mass index. Similarly, this review showed the behavioral and psychosocial effects derived from intervention programs, revealing in some cases improvements in the intention to be physically active [50], sedentary behavior [63], lifestyle [52], physical activity [51], attitude [68], and effort [44], and in others showing negative growth [73]. This fact may be due to the fidelity of the intervention where, due to the fact that most of the interventions that are developed, are carried out by teachers and non-researchers, the application of the strategies previously shown to the teacher is not applied in the intervention. In this sense, previous research used audio or video recording devices where this fidelity is ensured [54,62,63]. Regarding the effects on psychosocial variables, in some cases improvements in well-being were shown [43], self-efficacy [44,46], appearance [57], psychological condition [68], subjetive norms [55], psychosocial climate [70], and prosocial and antisocial behavior [47]. In this regard, the changes in the psychosocial variables could be due to the intervention procedure developed by the different researchers where the results can guide the theoretical developments and improvements, which then influence the subsequent research and practice in a significant way [83]. It is also important to point out the effects in PA after an intervention, as in many of the investigations, it was the means to improve the psychosocial variables [46,51,77].

Finally, in relation to the intervention time, the result showed that time is an essential variable for changes to occur. In this sense, the interventions with a duration of less than three months [41,66,73] presented fewer effects than those whose duration ranged from three months to a year [22,49,50,52,54,58]. However, there are exceptions where shorter interventions showed significant effects in most of their variables. This may be due to the methodology used or the professional who performed the intervention [46,51,77]. In this sense, Texeira et al. [37] highlighted the need for interventions greater than 3 months and adds that there are a limited number of interventions in the PA domain, and they are highly varied. Nevertheless, they concluded that autonomous motivation might change behaviors and maintain them over time. However, the work of Fu et al. [51] was based on the application for nine weeks of the Sport Play and Active Recreation for Kids program (SPARK) to increase PA in schoolage children. In this work, pedometers were used, which could favor the increase in PA through self-informed feedback. In this regard, Bronikowski et al. [46] also used pedometers (Garmin vivofit), on which participants had to register and upload data once a week. Shannon et al. [77] conducted an intervention based on healthy choices, which used techniques from the Self-Determination Theory along with resources that were sent to parents for their collaboration. Furthermore, interventions that lasted more than a year showed stagnation in the rating of their measures [22]. This fact may be explained by the teachers’ difficulty to motivate, capture the students’ attention, and maintain it over time. In this sense, an excessively short time does not produce effects, and an excessively long time (more than a year) can produce a decrease in the levels of motivation due to the absence of the element of novelty.

## 5. Conclusions

The results of this systematic review should be interpreted with caution due to the variety of effects analyzed and the methods used. Some of the weaknesses of the studies include a temporal limitation, which makes it difficult to obtain the effects of the intervention. In addition, some of the investigations did not include control guidelines to confirm which teachers and technicians applied the strategies and techniques previously taught by researchers. More intervention studies are required to expand on other motivational theories in order and to assess the effects on variables over time. An important limitation of the present review is that it did not include studies using a qualitative methodology. The absence of these studies precluded knowing testimonies transcribed from other interventions.

Future studies should be directed towards the complementation of the present systematic review with a meta-analysis that assesses the size of the effect of the interventions. In addition, future researchshould include mixed methodologies in their interventions and implement new strategies that will have a greater effect on the participants. In this sense, it is important to take advantage of new technologies to improve collaboration among teachers, families, and peers in order to carry out more effective interventions whose outcomes are maintained over time.

In conclusion, this review of the interventions carried out in the educational context provides a series of methodological recommendations when proposing and developing future intervention studies. In this sense, it is of utmost importance to highlight the strategies and the temporal nature of the studies that did maintain significant changes over time. Moreover, new future lines of research are also glimpsed. First of all, a new research trends in the female gender are put into focus as can be seen in the following research [71], which could lead to the significant decline of the female gender PA levels during adolescence. Secondly, it also highlights web-based interventions as a way to reinforce behavior. And finally, it is important to improve school PA with active breaks and brain breaks (mental breaks).

## Figures and Tables

**Figure 1 ijerph-17-00999-f001:**
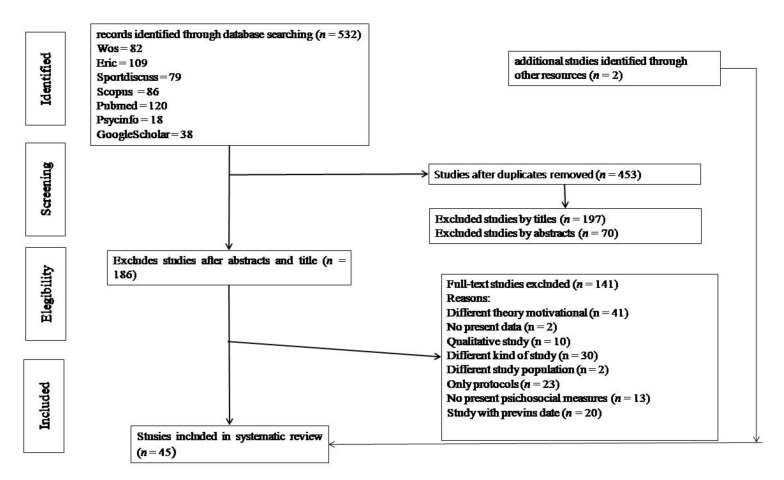
Flowchart of items through the search process.

**Table 1 ijerph-17-00999-t001:** Study selection criteria.

PICOS	Eligibility Criteria
Population	Children and adolescents between six and eighteen years old
Intervention	Intervention studies based on motivation for the promotion of physical activity and psychosocial benefits
Comparison	Compare an intervention group based on motivational principles that promotes physical activity through different strategies with a control group
Outcome	Any physical or psychosocial outcome is measured or reported.Psychosocial outcomes: individual’s social and psychological aspects including, but not limited to cognitions, emotions, and mental health

**Table 2 ijerph-17-00999-t002:** Quality of studies.

Authors (Year) [Reference]	a	b	c	d	e	Total Score	Quality Level
Abós et al. (2016) [41]	1	2	2	No	Yes	5	HQ
Amado et al. (2014) [21]	1	1	2	No	Yes	4	MQ
Amado et al. (2017) [42]	2	2	2	No	Yes	6	HQ
Babic et al. (2016) [43]	2	2	2	Yes	Yes	6	HQ
Bechter et al. (2019) [44]	2	2	2	Yes	Yes	6	HQ
Bortolí et al. (2015) [45]	2	2	2	No	Yes	6	HQ
Bronikowskiet al. (2016) [46]	2	2	2	Yes	No	6	HQ
Cheon et al. (2016) [22]	2	2	2	Yes	Yes	6	HQ
Cheon et al. (2018) [47]	2	2	2	Yes	Yes	6	HQ
Cuevas et al. (2016) [48]	2	2	1	Yes	Yes	5	HQ
Fernández-Rio et al. (2016) [49]	2	2	1	Yes	Yes	5	HQ
Franco et al. (2017) [50]	1	2	1	No	Yes	4	MQ
Fu et al. (2016) [51]	2	2	2	No	N/a	6	HQ
García-Calvo et al. (2015) [23]	2	2	2	No	No	6	HQ
Girelli et al. (2016) [52]	2	2	2	Yes	No	6	HQ
González-Cutre et al. (2014) [53]	1	2	1	No	N/a	4	MQ
González-Cutre et al. (2016) [54]	2	2	2	No	Yes	6	HQ
Grasten et al. (2015) [55]	2	2	1	No	No	5	HQ
Grasten et al. (2019) [56]	2	2	2	No	No	6	HQ
Hajar et al. (2019) [57]	2	1	1	Yes	N/a	4	MQ
How et al. (2013) [58]	2	2	2	Yes	No	6	HQ
Kanlayane et al. (2017) [59]	2	2	1	Yes	Yes	5	HQ
Kokkonen et al. (2018) [60]	2	1	2	No	Yes	5	HQ
Laroche et al. (2019) [61]	2	2	2	No	No	6	HQ
Lonsdale et al. (2013) [62]	2	2	2	Yes	Yes	6	HQ
Lonsdale et al. (2017) [63]	2	2	2	Yes	Yes	6	HQ
Lubans et al. (2016) [64]	2	2	2	Yes	Yes	6	HQ
Nicaise et al. (2014) [65]	1	2	2	N/a	N/a	5	HQ
Palmer et al. (2018) [66]	2	2	1	No	No	5	HQ
Perlman et al. (2013) [67]	2	2	2	Yes	Yes	6	HQ
Piipari et al. (2018) [68]	2	2	2	Yes	Yes	6	HQ
Quaresma et al. (2014) [69]	2	2	2	Yes	N/a	6	HQ
Riiser et al. (2014) [40]	2	2	1	No	N/a	5	HQ
Rhodes et al. (2018) [70]	2	2	2	Yes	Yes	6	HQ
Robbins et al. (2019) [71]	2	2	2	Yes	N/a	6	HQ
Rokka et al. (2019) [72]	2	1	1	No	No	4	MQ
Sánchez-Oliva et al. (2017) [73]	2	2	2	Yes	No	6	HQ
Sebire et al. (2016) [39]	2	2	2	Yes	Yes	6	HQ
Sevilet al. (2015) [74]	2	2	1	No	No	5	HQ
Sevil et al. (2016) [75]	2	2	2	Yes	No	6	HQ
Sevil et al. (2018) [76]	2	2	2	Yes	No	6	HQ
Shannon et al. (2018) [77]	2	2	2	No	No	6	HQ
Smith et al. (2017) [78]	2	2	2	Yes	N/a	6	HQ
Stephen Nation Grainger (2017) [79]	1	2	1	Yes	N/a	4	MQ
Tilga et al. (2019) [80]	2	2	2	Yes	Yes	6	HQ

Note: Range for the total score: high quality (HQ); medium quality (MQ); low quality (LQ). Parameter A: number of participants: 0 = less than 10; 1 = from 10 to 50 participants; 2 = more than 50 participants.Parameter B: description of the population and variables: 0 = fewer items than those required for value 1; 1 = sex, age, health status, and physical level; 2 = more items than those required for value 1. Parameter C: statistical analyses included in the study: 0 = such analyses not included in the value 1; 1 = error indexes or regression analysis; 2 = at least 3 graphical elements or repeated measures ANOVA. Parameter D: is the studie randomized? Parameter E: fidelity in the implementation.

**Table 3 ijerph-17-00999-t003:** Characteristics of study design and sampling.

Author (Year) [Reference]	Objective	Participants	Variables	Intervention Time	Data Sources	Results	Conclusions
Abós et al. (2016) [41]	To evaluate the effectiveness of applications of strategies based on TARGET, and the predisposition towards PA	35 students 15–17 yearsold	Climate scale of perceived motivation. Questionnaire to assess the BPN support Predisposition towards PA	The intervention was based on 12 acrosport lessons through the support of BPN. A first measure was taken at the beginning of the unit and two weeks after the end of the lessons. 2 teachers were instructed throughout 60h so that the intervention program was correctly fulfilled	Quasi-experimental design with non-equivalent groups.	Tests of normality, descriptive analysis and MANOVA were carried out	The importance in the learning of the task orientation in the motivation The importance of support for BPN can increase the predisposition towards PA
Amado et al. (2014) [21]	Verify the effect produced on the motivation of PE students of a multi-dimensional programme in dance teaching sessions	47 students from 14 to 16 years old	Self determination level BPN	12 teachin session, with two weekly of 50 min session throught week	Quasi-experimental design with non-equivalent groups	Descriptive analysis ANOVA and MANOVA were carried out	The programme’s usefulness in increasing the students’ motivation towards this content, which is so complicated for teachers of this area to develop
Amado et al. (2017) [42]	Find out gender differences in motivational processes, affective consequences, and behavior after a dance education program	12 teachers 921 students from 11 to 17 years old	Perception of support from the BPN. Perception of satisfaction of the BPN. Levels of motivation Utility. Enjoyment and effort Positive behavior	Intervention program covered a total of 35H, 12 sessions of a didactic unit	Experimental design CG and EG, in moments of pretest, posttest,	Descriptive, MANOVA, and Analysis of the variance of three factors, with two intersubject factors (sig differences in gender)	A creative methodology in the development of content increases motivation
Babic et al. (2016) [43]	Evaluate the impact of the “Switch-off 4 Healthy Minds” (S4HM) intervention onrecreational screen-time in adolescents.	322 adolescents from 12 to 14 years old	Sedentary activity PA Motivation to limit screen time BMI Psichological well-being Distress Physical self-concept	Interventionprogramover 6 month	Cluster randomized control trial with CG and EG	Descriptivesanálisis Mediationanalysis	The present trial was ineffective in its primary aim of reducing recreational screen-time. Significant intervention effects were observed for participants” autonomous motivation to limit screen-time, which mediated changes in screen-time
Bechter et al. (2019) [44]	Implement a teacher training program—based on well-established pedagogical strategies to improve key student outcomes (e.g., PE motivation, self-efficacy)	554 hightschools students from 12 to 16 years old, and 19 PE teachers	Teachers learning strategies Motivation BPN satisfaction Effort PE learning efficacy	Intervention program was based on two time of measures (i.e., initial measure and follow up) and the intervention was developed in five week with teacher learning strategies	Experimental design CG and EG, in moments of base line and follow up	Descriptive statistics MANOVA, Linear mixed models	Findings demonstrate that teacher training programs targeting the use of student-centered teaching strategies may be beneficial for promoting desirable motivational outcomes, and provide insight into the mechanisms responsible for these positive in-class effects
Bortolí et al. (2015) [45]	Examine the effect of motivational climate manipulation on students’ perception of climate, taking into account the effect of individual goal orientation.	108 female students from 14 to 15 years old	Perceived motivational climate Goal orientations Psychobiosocial state	We conducted three 2-h seminars with teachers before the data collection and intervention, as well as three meetings during the treatment that lasted approximately 40 min each. Each teacher was responsible for a “group involved in the task” and a “group committed to the ego”	The design was 2 × 2 2 groups during two times and the intervention was for 6 weeks, twice a week	Analysis of descriptive statistics, and ANCOVA of repeated measures	Findings highlight the link between several aspects of the multidimensional experience related to emotion in PE, in which the emotional and motivational contents are consistent with the content of performance
Bronikowski et al. (2016) [46]	To assess the effectiveness of two different goal orientations for the increase and maintenance of PA in adolescents	65 adolescentsof 17 yearsold	PA Support for teachers and classmate Self-efficacy	An intervention was developed in 2 different intervention groups, each with different strategies. 8 weeksduration	An experimental design was used, Pre-test start and posttest at the end	Descriptive statistics and wilcoxon test and Mann Whitney U were used, ANOVA and Post hoc test were used	The teacher’s support is more effective, than the objectives and strategies for the achievement of improving the MVPA
Cheon et al. (2016) [22]	Test an intervention by the teacher through a more autonomous teaching style in detraction from a more controlling style that promotes BPN	1017 students 19 teachers	Support for teacher autonomy Support for the teacher’s style of control BPN satisfaction and frustration Commitment in the class and amotivation	19 teachers, an instruction of 2 h and 30 to introduce the teaching of autonomy support. After 2 h focused on the development of specific skills in autonomous learning, part 3 of the instruction was through 2 h of discussion, they shared their experiences in the development of autonomy, the intervention was 8 months	CG (10) and EG (9) in 4 different times	Normality tests, descriptive statistics, correlations, *T* test, analysis of repeated measures	The satisfaction of BPN was increased through a teacher-centered intervention
Cheon et al. (2018) [47]	Implement an intervention program based on support for the autonomy of PE teachers to promote prosocial behaviors in students	33 teachers y 1824 students	Scoring ratios of teacher motivation styles Perception of prosocial and antisocial behavior according to the teacher Students’ perception of the teacher’s support for autonomy and style BPN satisfaction of students Prosocial and antisocial behavior of students Perception of deception	3 measures collected in the participating teachers: the first and second 2 weeks before starting the course, and the third during week 6 of the intervention. The intervention of the students took 19 weeks	15 teachers and their students’EG, and 18 teachers and their students CG	Descriptive; *T* Test; Multilevel analysis of repeated measures and finally calculation of the effect size; Structural equation model	This intervention has benefits for students with commitment, learning, motivation and psychosocial behaviors
Cuevas et al. (2016) [48]	Find out the impact of a sports education program on PE students regarding the levels of motivation	86 participants from 15 to 17 yearsold	Regulation of motivation. BPN thwarting Enjoy and boredom Intention to be physically active	19 lessons of a volleyball didactic unit (2× week)	Experimental Design CG and EG. CG traditional class and EG sports education program	Descriptive and ANOVA	No significant changes were found, although the changes in intrinsic motivation based on sports motivation should be highlighted
Fernández-Rio et al. (2016) [49]	Assess the impact of a cooperative learning program on student motivation	249 students, and 4 teachers in 4 different schools from 12 to 16 years old	Motivation Cooperation Perception of the students (open-ended question)	16-week intervention program (2 h per week) based on instruction to teachers through seminars	A design based on pretest-posttest, quasi-experimental, based on comparisons	Descriptive analyzes, Wilcoxon test, and *T* tests	The study showed that an intervention based on cooperative learning can increase the most self-determined types of motivation
Franco et al. (2017) [50]	Evaluate the effects of an intervention based on the support of BPN and the satisfaction of these needs	53 stundets from 12 to 15 yearsold	BPN Intrinsic motivation Intention to be physically active Enjoyment	A pre-post quasi-experimental design with two groups through 24 sessions (3 months), 10 h in strategies (experimental teacher training) was used	Intervention of 3 months, 2 measures (pre-post) 2 groups (CG and EG)	Kolmogorov test to assess the normality of the data. And the Mann Whitney U to evaluate differences	The usefulness of an intervention that incorporates strategies aimed at supporting BPN in satisfaction with autonomy and competence, and intrinsic motivation
Fu et al. (2016) [51]	To assess the effect of the SPARK program on levels of PA, cardiorespiratory endurance and motivation. In addition, evaluate the differences with respect to gender	175 students from 6th and 7th grade (82 boys, 92 girls)	PA in the classes Cardiorespiratory resistance Perceived competence Enjoyment	9-week program; 2 measurements 1 pretest and one posttest. The pedometers were used at the beginning (1st week) and also during the last week	CG and EG experimental design, in pretest and posttest moments	Descriptive statistics. *T* Tests and Analysis of covariance (ANCOVA)	The program demonstrated the effects on PA and motivation
García-Calvo et al. (2015) [23]	Measure the effects of a multidisciplinary intervention with teachers in the development of positive behaviors in PE classes	20 teachers and 777 students between 12 and 16 years	Positive behavior Positive behavior support	The entire study was developed from September 2011 to April 2012, but the intervention period was carried out from January to March	A quasi-experimental design with a CG and three groups (training program group, didactic unit group integral training group)	Descriptive statistics of each variable, pretest-poles, and the performance of a MANOVA and ANCOVA	The effectiveness of the intervention was demonstrated through the promotion of positive behaviors during PE classes
Girelli et al. (2016) [52]	Test the effectiveness of an intervention based on the Self-Determination Theory developed in Italy to promote healthy lifestyles in terms of PA and eating habits	866 participants, from 6 to 11 years old, 23 classes and 10 schools	Attitude toward PA Motivation towards PA Moderate daily exercise Daily sedentary activities Attitude towards the intake of healthy food. Motivation towards PA, daily consumption of healthy food. Daily snack consumption with high calories	9-month intervention with a minimum of 2 h per week, the CG only received one food seminar. Teachers and instructors were previously trained	Quasi-experimental design, with 2 Pre and Post measurements in a CG and a EG	Descriptive and exploratory statistics, ANOVA, correlation and intervention analysis were performed	The effects of the intervention program were demonstrated on a temporary basis, improving lifestyle and PA through healthy eating
González-Cutre et al. (2014) [53]	Analyse the effects of a school-based intervention to promote PA, utilising the postulates of the trans-contextual model of motivation	47 elementary schools students from 11 to 12 years old	Autonomy support Motivation in PE Motivation to leasure time Theory of planned behavior PA	Five week intervention. The intervention programme was conducted through videos with two session of 50 min per week	Quasi-experimental design, with 2 Pre and Post measurements in a CG and a EG	Non-parametric test for independent samples (Mann–WhitneyU) Differences with non-parametric test for related samples (Wilcoxon)	The study shows that this type of intervention couldallow the students to identify the benefits of PA and to integrate it intotheirlifestyle, thus increasing PA during their leisure time
González-Cutre et al. (2016) [54]	Assess the effect of a multidimensional intervention program to promote PA based on the Self-Determination Theory	88 students between 14 and 17 years old	Autonomy support BPN Types of motivation Physical self-concept Levels of PA	6-month intervention, data collection in 4 moments, before, after the first didactic unit, end of program (6 months) and follow-up (12), the questionnaires were filled in two different sessions	CG and EG × 4 different times	Normality tests, descriptive statistics, Man Withney U Multiple comparisons with the Wilcoxon test Size effect with Cliff delta	The effectiveness of a multidimensional intervention for the promotion of PA at school was demonstrated, however 6 months after the intervention the effects were lost
Grasten et al. (2015) [55]	To evaluate the effectiveness of an initiation program in the school of PA during a year	847 students from 12 to 14 years old	Motivational climate Goal orientations Auto report of moderate vigorous PA	Teachers were prepared through 4 workshops. To then intervene on the students. Theinterventionprocesswasduringoneyear in total.	Experimental design CG and EG, in pretest and posttest moments	Descriptive statistics. *T*-Tests and Analysis of covariance (ANCOVA), structural equation modeling	The findings of this study suggest that exposing students to additional PA throughout the school days and providing access to equipment and facilities during recess and breaks may be the most tangible way to increase their PA
Grasten et al. (2019) [56]	To examine the effectiveness of a programm focused on increasing mvpa and PE enjoyment	661 students from 11 to 13 years old and 46 teachers	Motivational climate MVPA (Prochaska) PE enjoyment	Teachers were prepared through 4 workshops. Interventions were made academic breaks giving greater autonomy with respect to facilities and materials	Longitudinal study (two years) with CG and EG in three measures	Descrptive statistics. Correlations, *T*-Test and path model	the MVPA did not increase and the enjoyment remained over time
Hajar et al. (2019) [57]	measure the efects of brain breaks on motives of participation in PA among primary school children	335 students from 10 to 11 years old	Demographic information enjoyment, mastery, competition ego, appearance, afiliation, physical condition, and psychological condition (PALMS-Y)	brain breaks activities, five minutes, five times per week, spread out for a period offourmonths	Cuasi-Experimental disign CG and EG, in pretest and posttest moments	Descrptivestatistics.Mixed factorial analysis of variance (ANOVA)	Brain breaks is successful in maintaining students’ motives for PA in four of the seven factors (enjoyment, competition, appearance, and psychological condition)
How et al. (2013) [58]	Assess an intervention in Pe through the election and see the effect of this on motivation and PA	257 students (137 intervention and 120 CG)	Motivation towards PE Levels of PA (accelerometers) Learning climate questionnaire	The intervention lasted 15 weeks. One week before the beginning of the study the teachers (4) received 40 min of explanation, all the students were assigned an intervention number. And 2 optionsofchoiceforthestudentswerepresented.	The data presented the following design: CG and EG and two data collection points	Descriptive analyzes, correlations, repeated measures anova, *T* tests	A system of options in PE classes can encourage support for autonomy and levels of PA in class
Kanlayanee et al. (2017) [59]	Assess the effectiveness of a weight reduction program through an improvement in BMI and autonomous motivation	304 participants	BMI Anthropometric measures Autonomous motivation for the exercise Dietetic self-regulation Autonomous Motivation of PA Autonomous motivation of dietary intakes	24-week intervention program, measures taken at the beginning, week 12 and week 24	Experimental Design: CG and EG	Descriptive, *T* tests, simple and multiple linear regression	The intervention program did not have a sufficient effect on BMI and autonomous motivation
Kokkonen et al. (2018) [60]	Assess the effect of an intervention based on creative PE	382 students from 9 to 11 years old	Motivational climate in PE Motivation towards PA in leisure time PA in general	It was based on two methods of data collection (web, and questionnaire), the methodology was presented in 2 seminars, later some teachers volunteered, CG teachers only focused on the motor goals	A quasi-experimental design with two groups and two data collection points	Normality tests, descriptive statistics, correlations, analysis of covariance, and a structural equation modeling	The intervention had a positive effect on PA in general and suggests that this intervention can increase the motivation of PA in leisure time
Laroche et al. (2019) [61]	Evaluate the direct relations between NPB, Autonomous motivation in a context of fitclub in female population	259 participantsfrom 14–15 yearsold	BPN Autonomous motivation Perceived behavioral control. Subjetive norms Intention PA leisure-time attitude	Two data measurements were collected, the first on the last day of the 8 weeks and the next three weeks after finishing	Longitudinal and quasi-experimental design, 1 EG in two data collection points	Descriptive, Correlations, Pathmodel	It is suggested that PA programs designed for adolescent girls should focus on improving autonomous motivation through opportunities for the development of BPN
Lonsdale et al. (2013) [62]	Promote PA	4 teachers, 16 PE classes, and 288 students	PA Sedentary Support for autonomy Motivation BPN satisfaction	Teachers from different schools were prepared to choose different types of practice during PE classes in one year	3 of the schools were experimental and 1 control; Experimental design with different moments of data collection: at the beginning, at the end and over time	Descriptive statistics, MANOVA and ANCOVA were used. In addition, a mixed analysis model was developed	The promotion of choice in the intervention provided short-term benefits in PA and in the reduction of sedentary behavior
Lonsdale et al. (2017) [63]	Test the effectiveness of an intervention in internet-based learning	1421 students from 14 schools	Demographic and anthropometric information MVPA through accelerometers Motivational mediators Sedentary behavior	The intervention was based on the learning of the teachers of strategies that improve students’motivation with the use of Internet and electronic resources	Three measures, baseline, post intervention, maintenance of the intervention.	Mixed linear models were used	Increase the level of moderate vigorous PA for adolescents
Lubans et al. (2016) [64]	To assess the impact of the ATLAS program on the prevention of obesity in children and adolescents	361 adolescents aged 12–14 years	Adiposity, PA, sedentary behavior, consumption of sweets, fitness, motivation and skill-competence in resistance training	20-week intervention that improved school sports programs, through interactive elements, parents, limiting time on the screen	Experimental design CG and EG, in moments of pretest, posttest, and post intervention measured 18 months later	Descriptive and Mixed Model which evaluated the impact of the group	The intervention was successful in producing positive effects but did not affect its objective of preventing obesity. Similarly, itshowedthegreatpotentialofschoolinterventions
Nicaise et al. (2014) [65]	Evaluate the effect of a PA program in relation to the Muslim religion	46 students aged 11–13 years, 43 of them completed all the measures	PA Enjoyment Motivation	The intervention lasted 8 weeks and one of the measures was at 2 weeks and another at 8 weeks	1 group with pre and post measures	Descriptiveanalysis and MANOVA	This program improved the enjoyment of PA
Palmer et al. (2017) [66]	Examine the effect of a program based on a didactic unit on 5 correlations in PA	300 students of grade 7 (12–13 years old)	PA, motivation towards PA, perceived sports competition, self-efficacy to eliminate barriers, beliefs of sports ability, perceived physical environment	4-week intervention based on a modified mountain bike unit	Intervention of 4 weeks with 3 groups and three measures pre, post, and follow-up	Descriptive analyzes, ANOVA, chi square	The results did not show main effects in the intervention, although they suggested that PE can influence certain correlations and an autonomous motivation for PA
Perlman et al. (2013) [67]	Assess the psychosocial, motivational and affective responses of students in two different learning contexts	41 EG students and 38 CG (9–10 years old)	BPN support Self-determined motivation Affected Enjoyment	Each class participated in a basketball unit during 4 weeks (16 lessons)	All the lessons were recorded on video and analyzed by a systematic observation system designed by Sarrazin et al. (2006) The systematic observation tool codifies the specific interactions between teachers and students in 15 categories	Descriptive statistics Repeated measurement anova	In general, current findings reinforce the relevance of self-determination within PE and the applied benefits associated with teaching PE using a highly autonomous learning context
Piipari et al. (2018) [68]	Evaluate the impact of an autonomous teaching program on students.	408 students, 11–13 years old, 19 PE classes, and 8 teachers	Support for perceived autonomy Support for the motivation perceived in the PE Autonomous motivation perceived in the exercise Intention to be physically active PA	CG and EG with two different treatments. Three phases of data collection Base line; week 4; and one week after the intervention. The teachers were trained (3 h) how to support autonomy. 8 week intervention	Quasi-experimental design with non-equivalent groups and three moments of data collection	Descriptive Correlations Analysis of covariance Structural Equation Modeling	The evidence that has the support for autonomy in motivation in PE is shown
Quaresma et al. (2014) [69]	Assess the effects of social support and regulation of exercise behavior in PA and quality of life	1042 students from 10 to 16 yearsold	PA Quality of life Motivation Parental and peers support.	The intervention program was developed during 24 months, an increase of 90 min per week, the health and weight sessions were carried out, emphasizing the knowledge of PA and feeding. Theinterventionwasmadebytheteachers	A study of two measures with CG and EG	Descriptive analyzes General lineal model and model of multiple mediators	An increase in parent and peer support in motivation through exercise represent mechanisms associated with high levels of PA and quality of life
Riiser et al. (2014) [40]	Evaluate the effectiveness of an internet-based primary care intervention	120 participants 84 CG and 36 EG 13–15 years old	Cardiorespiratory fitness. Quality of life. PA Self-determined motivation Body image Body mass index	12-week intervention with three measures: 1st before the intervention, 2nd immediately after, 3rd measurement 1 year after	Experimental Design: CG and EG	Means, minimums and maximums were used and compared with non-parametric tests, and *t* test, and the effect was calculated through D Cohen	The effectiveness of an Internet-based intervention for the improvement of respiratory fitness and quality of life is demonstrated
Rhodes et al. (2018) [70]	Compare the effect on an exergame intervention (exergame bike, standard bike) among children on motivational variables	73 insufficiently active children from 10–14 years old	Affective attitude (enjoyment, boredom) Instrumental attitude (utility, beneficial) Motivation towards the exercise Subjective norms, Perceived behavior	13 weeks intervention with thre measures, 1 basseline, 2 (six week), and 3, 13 weeks	Two arm paralel design with EG and CG	Descriptives analyzes, correlation, and path model	Unique exposure research designs do not They accurately reflect the motivations for the long-term exercise game. Also the role of parents can favor an attitude
Robbins et al. (2019) [71]	Assess whether constructs derived from the practice of PA are mediated by SDT and health promotion models for a population of adolescent girls	1519 girls from 12 yearsold	Demographic data (age, etnicithy, race) Perceived benefits scale of the PA Scale of perceived barriers to PA Social support Self-efficacy Enjoy and motivation	17-week intervention with two measure 1st immediately after 17 week, 2nd measurement 9 month after inervention	A study of two measures with CG and EG	Using R statistical software, linear mixed-effects models and path analysis	Enjoyment of PA and social support for PA may be important mediators of MVPA in underserved young adolescent girls
Rokka et al. (2019) [72]	Investigate the effect of a dance aerobic intervention program on intrinsic motivation and perceived motivation climate in schools students	160 healthy students from 12 to 13 years old	Intrinsic motivation Enjoyment Perceived motivational climate	10 week intervention (two sesión per week) with two measuremens (i.e., initial measure and final measure)	A study of two measures with CG and EG	Descriptive analyzes, Correlation, *T*-Test and ANOVA of repeated measure	The intervention programs involved in dancing are a good way to promote motivation, enjoyment through PE classes
Sánchez-Oliva et al. (2017) [73]	Analyze the effect of the intervention process to evaluate changes in motivational processes	21 teachers and 836 students from 12–16 years old	BPN support Satisfaction of the BPN Motivation Intention to be physically active	Teacher preparation program during 15 h. 3 sessions per week 10 weeks	Experimental design CG and EG, in moments of pretest, posttest, and post-intervention measure	Descriptive, Multilevel model and ANCOVA. Boys reported higher scores. 1st grade students scored significantly higher	Positive effects of an intervention program with teachers on the perceived need for support and autonomous motivation within PE clases
Sebire et al. (2016) [39]	Evaluate the intervention process through a mixed methodology	539 children	Motivation towards PA, BPN satisfaction, perception of support for autonomy, support for self-efficacy based on teaching	Intervention of 20 weeks, during the intervention 4 measures were taken, and another assemement 4 months after the intervention	Experimental Design: CG and EG	Descriptive, evidence Code-based transcription for interviews	The intervention was successful and showed that it can influence the motivation for PA. The study demonstrated the effectiveness of mixed methods to support motivation
Sevil et al. (2015) [74]	Test genders‘ influence on motivational variables and cognitive and affective consequences along different PE didactic units.	66 students from 15 to 17 years old	BPN Self determination motivation. Enjoy and boredom Predisposition toward the content	Intervention of four month, of 60 min with two sesión per week	Quasi-experimental design with non-equivalent groups	Normality analysis Descriptive analysis Manova	It is important that curricular contents are developed taking into account the gender variable. Therefore, it is important to develop the BPNs to generate more fun and willingness to exercise
Sevil et al. (2016) [75]	Evaluate the effectiveness of an intervention program in a series of motivational variables in a body language teaching unit	224 studentsfrom 12–14 yearsold	Motivational climate Satisfaction of the BPN Self-determined motivation Enjoyment/satisfaction and boredom	A program for the 55-h EG teacher was prepared according to the guidelines indicated by Braithwaite et al. 2011. In the second stage, a group of experts prepared the DU February-April, two sessions a week, 50 min per sesión	Quasi-experimental design with non-equivalent groups	Descriptive, Manova, and the effect size with the partial Eta Squared static	The results indicated the importance of creating a task-oriented climate in PE classes that will promote the most positive experiences
Sevil et al. (2018) [76]	Assess the effects of an intervention program in the interpersonal teaching style and a series of motivational variables and consequences present in the “bright side” and the “dark side” of the motivation	103 students	Support for BPN Motivational climate BPN in the field of the PE BPNES satisfaction EFNP thwarting Self-determined motivation Fun/Satisfaction and boredom Predisposition to content Defiantopposition	Intervention of 8 sessions plus 1 additional session of Didactic Unit (athletics). Teacher training of 30 h (2 teachers). 2 pretest and posttestmeasurements	Experimental Design CG and EG.	Descriptive, Manova, and effect size through Partial Eta Squared (*η*^2^p)	The intervention program is effective in both genders, promoting even better results in this content in boys
Shannon et al. (2018) [77]	Analyze the effect of a choice program on health through a model of autonomy support, BPN satisfaction, intrinsic motivation with PA and well-being	155 participants	MVPA (accelerometers) Wellness Screen time Support for autonomy NPB satisfaction Motivation (4 scales)	2 groups × 2 measures. The data was collected in week 0 and week 11 The intervention period was 10 weeks during the school period using 2 h and 15 total minutes per week	Experimental design CG and EG, in pretest and posttest moments.	Descriptive statistics were used and two mediating models (1 with PA and another with welfare)	The study showed that this intervention design improves the MVPA and the welfare through the improvement of support in the autonomy
Smith et al. (2016) [78]	Assess the effect of the program on the prevention of obesity	361 adolescents, 12–14 years old from 14 high schools and 2 teachers	Demographic data Screen time Motivation to limit screen time Screen time rules in the family home	School program of 20 weeks (3 s × 20 min of information from researchers to students, 20 × 90 min sessions of PA, 6 × 20 tutorials on eating habits, and news bulletins screen time to parents	Experimental design with different moments of data collection: at the beginning, at 4 months and at 18 months	Multilevelmediationanalysis	Increasing autonomous motivation to limit screen time can be a useful strategy to address this widespread behavior
Stephen Nation Grainger, (2017) [79]	Improve levels of physical exercise, using a PA monitor	10 studentsfrom 14–15 yearsold	Motivation PA through smart watch Qualitative method (interview)	It was based on a 6-week intervention, where the EG received feed-back after the PE class.	A study of two measures with CG and EG	Normality tests, *T* tests and correlations were used	A feed-back through technology can increase motivation in PE classes
Tilga et al. (2019) [80]	Interventions based on self-determination theory to help teachers support their students’ autonomy	321 students from 10–15 years old, and 28 PE teachers	Multidimensional autonomy-supportive behavior Multidimensional controlling behavior. Perceived need satisfaction and need frustration. Intrinsicmotivation	9 weeks intervention, where the EG has three measure (i.e., baseline, 4 weeks, follow up 9 weeks), and CG two measure (i.e., baseline and follow up)	A study experimental with CG and EG	Descriptive statics, Chi square, *T* tests, and ANCOVAS	This study provides initial evidence that WB-ASIP might increase the likelihood that teachers change their behavior, which, in turn, may facilitate students’ psychological needs satisfaction and motivation toward PE.

*Abbreviations*: PA, physical activity; BPN, basic psychological needs; PE, physical education; MVPA, moderate to vigorous physical activity; BMI, body mass index; CG, control group; EG, expiremental group; DU, didactic units; SDT, self-determination theory.

**Table 4 ijerph-17-00999-t004:** Outcome syntax by variable.

		Studies	
Theoretical Concepts	Intervention Variables	After Intervention Outcomes	Outcome in Control Group
Motivation	Intrinsic regulation	[48,49,50,53] Increasing to PE; [42,54,61,64,65,70,74,76,77,80]	[48,50] decrease to PE
	Integrated regulation	[54,61]	
	Identified regulation	[21,48,49,53] Increasing to PE; [42,44,54,61,64,65] Decreasing; [39,70,76,77,79]	[48,53]
	Autonomous motivation	[43,44,48,52] nonsignificant; [54,61,66,67,68,71,72,73] decrease; [40] no changes in motivation; [78] autonomous motivation for screen time	
	Interjected regulation	[42,53,64] decrease; [39] increase Experimental Group; [77] increase	[49,75] increase
	External regulation	[64] decrease; [65] decrease; [69] decrease; [77] decrease	
	Controlled motivation	[59] increase for dietetic intake	[63] decrease
	Amotivation	[22] decrease; [75] increased; [59] increased for PA and dietetic intake; [65,69] increased; [73] increased	[22] increasing; [75] increased; [54] increasing
Basic Need Satisfaction	Competence	[50,51] decrease; [42] in the female gender; [61,73] decrease; [39] decrease tendency; [74,76,77,80]	
	autonomy	[22,44,50,53] professor; parents and peers [54]; [42] in the female gender; [61,69] support for parents and peers; [73] decrease; [39] no changes; [74,76,77,80]	[53] professor; parents decreased, and peers increased
	relatedness	[42,44] female gender; [61,73] decrease; [39] decrease tendency; [74,77,80]	
	satisfaction	[22,47]	[22,47]
Basic Need Support	Competence	[42,66] perceived competence; [67,73] decrease; [76]	
	autonomy	[42,46,58,62] perceived; [67,68,73] decrease; [39,76,77]	
	relatedness	[42,67,73] decrease	
Basic Need Frustration	competence	[21,76] decrease; [80] decrease	
	Autonomy	[76] decrease	
	Relatedness	[44,48,76] decrease	
	Frustration	[22,47] decrease	[22,47] increase
Achievement Goal Theory	Task-oriented climate	[45,56,58] decrease [60,72]	[57,60] increase
	Ego-oriented climate	[45,55,56] increase; [72] decrease; [76] decrease	[60] increase
Behaviours	Intention to be physically active	[48,50] decreasing; [53,61,70]; [73] decrease; [76] predisposition towards content;	[48,53]
	Sedentary	[62,63,64] decrease screen time;	
	Lifestyle	[52] healthy intake	
	Physical activity	[43,46,51,53,54] PA in the spare time [55,56] decrease; [21,58,60] in the spare time; [61] in the spare time; [62,63] in the spare timeand MVPA; [65,66] during PE classes in the PE control group; [40,68]; [71] decrease [70,77] increase with accelerometers; [79] with smartwatch feed-back calories	[53] increase
	Attitude	[23,41,49,50,52] towards PA; [55,68] competition; [64] benefitsperceive; [70] affective andinstrumental attitude; [74] affective;	[23,55]
	Effort in PE	[23,42,44] Female; [72]	
	Behavioral perceived control	[23] self-control; respect [61]	
	Engagement	[22]	
AntropometricVariables	Cardiorrespiratory fitness	[40,51,64] changes showed	
	BMI/adiposity	[59] increase [64] slightly improved; [40] slightly reduction [39,78]	
PsichosocialVariables	Body image	[40]	
	Quality life	[69] no changes showed; [40] changes showed; [77]	
	Well-being	[43]	
	Funny/enjoy	[41,42,48,50,56] decrease, [57] decrease; [65,67,71,72,76]	[41]
	boredom	[48] decrease [76] decrease	[48] increased
	useldfulness	[42] only in female gender	
	self-efficacy	[39,41,44,46,66]	[66]
	appearance	[57]	[57]
	Psychological condition	[57,74] cognitive activity; [80] support to the cognitive perception	[57]
	Perceived barriers	[71] increased	
	Subjective norms	[53,61]	[53]
	Psychosocial climate	[45,57] affiliation; [76,80] perception of intimidation	
	Social support	[71,80] perception of organizational and procedural support	
	Prosocial behavior	[47]	[47]
	Antisocial behavior	[47] decrease; [76] defiant opposition, decrease	[47] increased

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
