# Peer review of "School-based Physical Activity Interventions in Children and Adolescents: A Systematic Review"

_ijerph, 2020, doi:10.3390/ijerph17030999_

Round 1

Reviewer 1 Report

There are several methodological flaws and weaknesses that should be properly amended and discussed. 

1) There are several typos (such as line 34, However, There, which should be However, there). In several points, English should be revised for the purpose of clarity and readability should be improved.

2) There is an inconsistent use of acronyms. At line 39, it is not the first time that physical activity is used (already mentioned at line 32), but the abbreviation occurs for the first time at line 39.  

3) Is the study protocol registered in PROSPERO?

4) Please list PICO/PECO criteria.

5) Search strategy does not appear to be comprehensive (in terms of relevant keywords).

6) The names of the bibliographic databases/thesauri are wrong: PsycINFO and not Psycinfo, SPORTDiscus and not Sport Discuss. Authors should clarifiy how they mined PubMed (via MEDLINE? via OVID?), etc. for the other databases.

7) Excluded studies with reason should be counted.

8) Outcomes can also be presented in synthetic tables.

9) Discussion should be expanded.

Author Response

Suggestion 1 Reviewer 1) There are several typos (such as line 34, However, There, which should be However, there). In several points, English should be revised for the purpose of clarity and readability should be improved.

Authors)We appreciate you rsuggestion for improvement and we have replaced “Thereforthere”. We have also decided to send the article to thetranslation service.

Suggestion 2 Reviewer 2) There is an inconsistent use of acronyms. At line 39, it is not the first time that physical activity is used (already mentioned at line 32), but the abbreviation occurs for the first time at line 39.  

Authors) Thank you for your contribution as it improves the quality of the article. In thissense, we have included the acronymon line 30, since it is the first time it appears. In the following times the acronym PA has been used.

Suggestion 3Reviewer 3) Is the study protocol registered in PROSPERO?

Authors) Dear editor, we understand the limitation of not having the PROSPERO review registered. In this regard, we say that we tried to register the systematic review in PROSPERO but it was rejected for the following reason: “Reviews that have started data extraction are not eligible for inclusion in PROSPERO. The aim of the register is to capture information at the design stage of a review.”

However, systematic reviews of relevant quality such as the following have not been registered in PROSPEROExercise, physical activity, and self-determination theory: a systematic review”  (Texeira et al., 2012); Self-determined motivation and physical activity in children andadolescents: A systematic review and meta-analysis (Owen et al., 2014); Associations between affect, basic psychological needs and motivation in physical activity contexts: systematic review and meta-analysis(Texeira, Marques y Palmeira, 2018); Efficacy of theory-based interventions to promote physical activity.

Similarly, we agree that not being registered in Prospero can be a limitation. However, similar reviews such as some that have been published recently in this journal do not have a prosperous record, and we do not doubt their high scientific quality.

Class Time Physical Activity Programs for Primary School Aged Children at Specialist Schools: A Systematic Mapping Review (Emonson et al., 2019) Active Commute in Relation to Cognition and Academic Achievement in Children and Adolescents: A Systematic Review and Future Recommendations (Phansikar et al., 2019)

Therefore, please do not consider the lack of the registration number as a criterion of poor quality, because for the reason mentioned above, in the first paragraph of this suggestion, we do not have it

Suggestion 4Reviewer 4) Please list PICO/PECO criteria.

Authors) We would like to thank reviewer ssuggestion because it improves the quality and meaning of the article. In this line, a PICO table has been added in which the study selection criteria are explained.

We also believe that this table complements the subsequent information that speaks of the inclusion and exclusion criteria that have been taken into account in the systematic review.

Suggestion 5 Reviewer 5) Search strategy does not appear to be comprehensive (in terms of relevant keywords)

Authors) Thank you for your suggestion for improvement. However, the search terms in the current systematic review have been discussed before their completion and the authors agreed to use these terms.

Thus, we think that the terms motivation or intrinsic motivation would encompass those studies that work on the motivational theories that are described in the theoretical framework, and on which the majority of intervention studies for the promotion of physical activity and other benefits are based of psychosocial nature. A part from these key terms, it was necessary to delimit those studies that included only the interventions, so we used the terms "Intervention" OR "Program" to collect all the works that have carried out intervention programs, the terms "PhysicalActivity" OR “PhysicalEducation” to delimit the context in which this review is centered AND “Children” AND “Teenegers” OR “Adolescents” in order to delimit those based on physical activity directed at the school context in children or adolescents. Therefore, we consider that this set of words must appear at least once in the works that are intended to be contemplated in this systematic review.

Suggestion 6 Reviewer) The names of the bibliographic databases/thesauri are wrong: PsycINFO and not Psycinfo, SPORTDiscus and not Sport Discuss. Authors should clarifiy how they mined PubMed (via MEDLINE? via OVID?), etc. for the other databases.

Authors) Thanks for the suggestions we have taken your input into account and modified it based on the correct names of the bibliographic databases. In addition, the doubt that the reviewer has with the PubMed database, which was extracted through MEDLINE, has been resolved.

Suggestion 7)Reviewer) Excluded studies with reason should be counted.

Authors)We appreciate reviewers´suggestion for improvement. In this sense, the recommendation has been followed, which has allowed a new review of the articles, and the flow chart has been updated, counting the reasons for the number of excluded studies.

In addition, five new studies have been added that were initially excluded. In this sense, all the excluded articles were read and we found articles that met the inclusion criteria, and that were not previously integrated into there view.

There as ons why they were included were that they were interventions based on one of the two motivational theories that conceptualize this systematic review, for a school population, which was based on the promotion of physical activity and finally the psychosocial benefits were valued.

Suggestion 8)Reviewer) Outcomes can also be presented in synthetic tables.

Authors)In response to your suggestion for improvement, a new table has been prepared where all the results are synthesized for each of the variables developed in the different interventions.

Suggestion 9)Reviewer) Discussion should be expanded.

Authors) We appreciate your suggestion for improvement. In this sense, the discussion has been expanded by reporting information on thee ffects of interventions on motivation and psychosocial variables.

To see the new version of the document click on the attached file.

Reviewer 2 Report

I find that there is a need for systematic reviews on PA promotion interventions aimed at school children in the school contexts. Hence, I commend the authors on the importance of the research project. The authors have searched an extensive number of relevant databases. However, I find the scope of the present study to be too wide and unclear. This limits the contribution of the review to the field. Also, I find the information provided about the research method to lack important aspects which I would expect from a thorough systematic review.

Research questions

The research questions are not clear, such as page 3, lines 103-104: What incidents have the interventions (…) on the curricular or extracurricular school context”. What do you mean by “incident”? Do you mean effects? And what part of the school context are you referring to? It is also unclear what you define as the primary and secondary outcome measures. Is PA sees as the primary outcome measure, and psychosocial benefits the secondary outcomes. A multitude of possible outcome measures are listed, and they vary across the manuscript. In the Abstract and Introduction section, the following are listed: psychosocial behavior, self-esteem, body image (the two latter I would define as attitudes and not primarily behavior), sedentary lifestyle, quality of life etc. In the Material and methods section, fun, enjoyment, boredom and psychosocial climate are included. At “3.9 Psychosocial effects”, self-concept, prosocial behavior, and utility is added to the list. The concepts and the research findings related to these concepts in general is poorly defined. With a multitude of concepts and limited number of studies, one should be cautious about concluding about the causal relationships between motivational processes, PA and the secondary outcome measures.

Materials and methods

According to the manuscript, the review adheres to the PRISMA statement. I would expect that a protocol was developed prior to data collection? The protocol should preferably be published at web-site for systematic review registration. Was this done? Quality assessment of intervention studies commonly includes 1) fidelity to the intervention protocol when delivering the intervention, and 2) the participants’ adherence to the intervention (did they received the dose needed to expect results). I think these important criteria should be included in the assessments of the study quality. The CONSORT statement and checklist are commonly used to assess quality. You refer to a review on fitness testing studies among young people (38). I think you should apply another framework for quality assessments developed specifically for intervention studies/RCT. Page 2, lines 46-47: you comment on the lack of studies with a quasi-experimental design. Could you specify why you focus on quasi-experimental studies and not RCT? The time period for data collection (search and scanning of papers) should be specified in terms of dates and number of weeks. In the abstract, it is stated that the search process was from June 2011 to September 2019, but I gather this was the publishing dates of the papers? Did you include any grey material such as research reports or unpublished manuscripts? This should be stated clearly The paper refers to statistically significant changes. It is fairly common to refer to other measures of effect such as effect sizes. I gather a large number of the studies included have measured the effect sized of the changes and the between-groups effects. These should be included to assess the strength of the findings, not just whether they were statistically significant. the studies may not use the exact same measure of effect. However, common measures of effect size tend to have cut-off values specifying whether the results would be regarded as small, moderate or large. Page 5, lines 172-174: you define adolescents as ages of 6-17. Do you mean 12-17? Page 15, lines : you state that most of the studies were quasi-experimental. This entails that the intervention was not randomly allocated to one group after baseline assessments, but rather that the group defined as the intervention group was decided on from the start or based on the results of the baseline assessments. Is this really the case for all the studies? In case they are in fact randomized, I think you also should state whether they were cluster og individually randomized.

Table 2 needs a more stringent presentation:

Intervention time – the information presented is a mix of duration (weeks), dosage the children received (hours) and training (of the providers of the intervention) in addition to some comments on measuring points and whether this is a pre-post measurement study. The distinction between intervention “development” and “carried out” is several places not clear. I would define the word “development” as the process of designing the intervention before it is actually delivered to the target population. Data sources – the majority of the information presented is actually about design, sometimes with additional information about the measurements (pre-post). I also think that information about randomization should be added (if it is a RCT and preferably if it is a cluster-randomized study) and whether the study had pre and post assessments or not What you present under the label “Results” is predominantly the analyses. The column “Conclusions” seems to mix of findings and recommendations drawn directly form the papers. I think more statistical measures of effects should be presented, preferably effect sizes.

Results

Page 16, lines 238-240: Here you mention the effects on the SDT based interventions on other primary outcome variables like calorie intake and screen time. I suggest you restrict your analysis to PA. Page 16, lines 230-231: What do you mean by the “values of motivation” Page 16, lines 266-267: You state that “most of the intervention studies showed a statistically significant increase in PA” and you list 5 study references. Then you write “some intervention studies did not…” and you list 6 study references. I would suggest that it’s the other way around, a majority of studies no not demonstrate an effect. In the Abstract, “the female gender as a participant of special interest” is mentioned. I would expect a more thorough discussion of the gender-effect in the manuscript.

Author Response

Suggestion 1Reviewer) Research question

The research questions are not clear, such as page 3, lines 103-104: What incidents have the interventions (…) on the curricular or extracurricular school context”. What do you mean by “incident”? Do you mean effects? And what part of the school context are you referring to? It is also unclear what you define as the primary and secondary outcome measures. Is PA sees as the primary outcome measure, and psychosocial benefits the secondary outcomes. A multitude of possible outcome measures are listed, and they vary across the manuscript. In the Abstract and Introduction section, the following are listed: psychosocial behavior, self-esteem, body image (the two latter I would define as attitudes and not primarily behavior), sedentary lifestyle, quality of life etc. In the Material and methods section, fun, enjoyment, boredom and psychosocial climate are included. At “3.9 Psychosocial effects”, self-concept, prosocial behavior, and utility is added to the list. The concepts and the research findings related to these concepts in general is poorly defined. With a multitude of concepts and limited number of studies, one should be cautious about concluding about the causal relationships between motivational processes, PA and the secondary outcome measures.

AUTHORS) Dear editor, with respect to the word incidence, we refer both to the effects they produce and to the presence that the theory of self-determination and achievement goals have in the interventions carried out in the school context, whether in the regular schedule or extracurricular

In relation to the suggestion you indicated regarding what the primary results are and which are the secondary ones, we appreciate your suggestion. In this sense, our objective is to assess the presence of those interventions that have been based on the theory of self-determination or achievement goals theory through the promotion of physical activity or physical education that promote psychosocial change, understanding as psychosocial changes to those who refer to mental health or social adjustment. Similarly, the psychosocial concept refers to the nature of human behavior in its social aspect or related to it. In this sense, relevant authors  (William and Deci, 1996;Ryan, Ntoumanis, Sallis, Standage) they use it to encompass behaviors and behaviors, such as those indicated in the text of the manuscript.

In relation to the suggestion about the concept of utility, the authors in this intervention refer to the instrumental attitude, which is composed of utility and benefit.

Moreover, we appreciate your suggestion and we feel that we could perceive that we were establishing causal relationships. In this sense we are not concluding with those causal relationships, where it is stated that the cause of the improvement of certain aspects is due to the application of the intervention program in the motivation and promotion of PA but that we systematically analyze its incidence ( effect), assessing whether the intervention carried out has served to improve any of the psychological aspects.

Suggestion 2Reviewer Materials and methods

According to the manuscript, the review adheres to the PRISMA statement. I would expect that a protocol was developed prior to data collection? The protocol should preferably be published at web-site for systematic review registration. Was this done? Quality assessment of intervention studies commonly includes 1) fidelity to the intervention protocol when delivering the intervention, and 2) the participants’ adherence to the intervention (did they received the dose needed to expect results). I think these important criteria should be included in the assessments of the study quality. The CONSORT statement and checklist are commonly used to assess quality. You refer to a review on fitness testing studies among young people (38). I think you should apply another framework for quality assessments developed specifically for intervention studies/RCT. Page 2, lines 46-47: you comment on the lack of studies with a quasi-experimental design. Could you specify why you focus on quasi-experimental studies and not RCT? The time period for data collection (search and scanning of papers) should be specified in terms of dates and number of weeks. In the abstract, it is stated that the search process was from June 2011 to September 2019, but I gather this was the publishing dates of the papers? Did you include any grey material such as research reports or unpublished manuscripts? This should be stated clearly The paper refers to statistically significant changes. It is fairly common to refer to other measures of effect such as effect sizes. I gather a large number of the studies included have measured the effect sized of the changes and the between-groups effects. These should be included to assess the strength of the findings, not just whether they were statistically significant. the studies may not use the exact same measure of effect. However, common measures of effect size tend to have cut-off values specifying whether the results would be regarded as small, moderate or large. Page 5, lines 172-174: you define adolescents as ages of 6-17. Do you mean 12-17? Page 15, lines : you state that most of the studies were quasi-experimental. This entails that the intervention was not randomly allocated to one group after baseline assessments, but rather that the group defined as the intervention group was decided on from the start or based on the results of the baseline assessments. Is this really the case for all the studies? In case they are in fact randomized, I think you also should state whether they were cluster og individually randomized.

Table 2 needs a more stringent presentation:

Intervention time – the information presented is a mix of duration (weeks), dosage the children received (hours) and training (of the providers of the intervention) in addition to some comments on measuring points and whether this is a pre-post measurement study. The distinction between intervention “development” and “carried out” is several places not clear. I would define the word “development” as the process of designing the intervention before it is actually delivered to the target population. Data sources – the majority of the information presented is actually about design, sometimes with additional information about the measurements (pre-post). I also think that information about randomization should be added (if it is a RCT and preferably if it is a cluster-randomized study) and whether the study had pre and post assessments or not What you present under the label “Results” is predominantly the analyses. The column “Conclusions” seems to mix of findings and recommendations drawn directly form the papers. I think more statistical measures of effects should be presented, preferably effect sizes.

AUTHORS) Dear reviewer, we appreciate your suggestions for improvement as they all contribute to the quality improvement of the manuscript. Then I will proceed to answer each of your suggestions for improvement in parts:

Regarding your suggestion about the protocol of the systematic review, tell you that we tried to register it in PROSPERO, but because of the phase in which we were in the systematic review (writing results) they did not accept the registration. In this line, we would like to indicate that the journal itself has published some recent systematic reviews that do not include PROSPERO registration code:

Class Time Physical Activity Programs for Primary School Aged Children at Specialist Schools: A Systematic Mapping Review (Emonson et al., 2019) Active Commute in Relation to Cognition and Academic Achievement in Children and Adolescents: A Systematic Review and Future Recommendations (Phansikar et al., 2019)

In relation to your suggestion to include the fidelity of the intervention protocol or its adherence, we have used the scale of Castro-piñero et al. (2010), which has been used in systematic reviews of high quality such as Pozo, Grao-Cruces and Perez (2017). However, if the reviewer considers that it is necessary to include it, we are willing to add it, since this information improves the quality of this systematic review.

At the time of selecting the articles of the present review, we did not exclude those who did not present a sample randomization process, since our objective was to assess whether interventions based on the motivational theories of Self-Determination or Achievement Goals had effects in the psychosocial variables. In the process of selecting titles, we considered all those who talked about effects, influence, incidence, importance and who were alluding to motivational and psychosocial processes, and the use of PA.

The article search period stipulates the date covered by this systematic review. Likewise, no gray matter is included in this review, only articles published during the dates stipulated in the review process.

With respect to the results of the interventions as indicated, we refer to statistically significant since it is a systematic review and not a meta-analysis. Similarly, if you consider it, we could add one more section with the effects of the interventions for each type of variable in a descriptive way. In this line, in relation to the effect, we think that the ideal, if a new section were included with the effects, would be the effects in relation to time and not effects between groups, since our goal is to value temporary changes. Logically, if an intervention is applied to a group, it will receive variation in one of the study variables, and the control group will maintain the same trend.

Table 2 shows a description of the interventions, which includes the objective, participants, variables, method, statistical analysis performed and conclusion. If the author considers it, new information could be added, however the length of this manuscript would be significantly extended. Similarly, if you wish we can add a supplementary table with the effects of each intervention from a temporal perspective and whether the intervention has been randomized or not, but the extension of the manuscript would be too long.

Suggestion 3Reviewer) Page 16, lines 238-240: Here you mention the effects on the SDT based interventions on other primary outcome variables like calorie intake and screen time. I suggest you restrict your analysis to PA. Page 16, lines 230-231: What do you mean by the “values of motivation” Page 16, lines 266-267: You state that “most of the intervention studies showed a statistically significant increase in PA” and you list 5 study references. Then you write “some intervention studies did not…” and you list 6 study references. I would suggest that it’s the other way around, a majority of studies no not demonstrate an effect. In the Abstract, “the female gender as a participant of special interest” is mentioned. I would expect a more thorough discussion of the gender-effect in the manuscript.

Authors) Dear reviewer, specifically for that study we refer to the use of motivation to eat healthier and to reduce screen time. In relation to the expression “motivation values” we refer to the participants' scores on motivation (increase or decrease)

Regarding those studies that showed changes in BP, a new results table has been added that lists the studies which improved BP and highlights those where BP decreased.

The objective of the discussion has not been to establish the presence of the female gender in the interventions and its process of change, but we have highlighted a current trend of the interventions. In this sense, as we mentioned earlier if you consider it we would add it. However the extension of the article would increase significantly.

In general terms, we consider that its recommendations for improvement provide great quality to the document. However, we ask that you answer us after looking in depth the new extension of the manuscript, and if you wish we apply the aforementioned changes on RCT, information in the tables, and effects of the intervention

Response general to reviewer

Finally, We (the authors) think after reviewing the length of the manuscript, and the information shown we believe would be sufficient. However, according to the suggestions of the reviewer if in a subsequent review he considers that the information regarding the effects, quality criteria ... etc.

We would agree to add it, as long as it is taken into account that the length of the document would increase significantly

To see the new version of the document click on the attached file

Reviewer 3 Report

The manuscript presents an importante theme for physical Activity with children and adolescents, and its importante to describe this information and make avaiable for readers in these áreas of expertise. It has interesting and very recent references, so it should be publish. 

Author Response

Dear reviewer, we welcome your suggestions and contributions, as you encourage us in this review process. However, we inform you that, given the suggestions for improvement by the rest of the reviewers, some aspects have been modified to add more quality to the manuscript. We hope that these modifications are liked by the reviewer.
Next we add the new version of the manuscript as an attachment.

Round 2

Reviewer 1 Report

The manuscript has been considerably improved and can now be accepted for publication.

Author Response

Dear reviewer, we appreciate your suggestion for improvement as it helps to increase the quality of the manuscript. In this regard we have followed his recommendation and the text has been corrected again by a native expert in the field.

Reviewer 2 Report

I find that the revised manuscript has not been substantially improved, the authors' comments reflect an insufficient knowledge on intervention design and quality assessments. The authors' seem in general rather reluctant to make the changes I recommend and do not seem to share my concerns. One example is the numerous variables included in the study, both mediating, primary and secondary outcome measures. This limits the contribution of the study in it's present form to the field of physical activity and SDT/AGT based intervention studies.
